# Effect of Nanosilica on the Undrained Shear Strength of Organic Soil

**DOI:** 10.3390/nano15090702

**Published:** 2025-05-07

**Authors:** Carlos Solórzano-Blacio, Jorge Albuja-Sánchez

**Affiliations:** 1Multidisciplinary Engineering Research Hub (MER Hub), Faculty of Habitat, Infrastructure, and Creativity, Pontificia Universidad Católica del Ecuador (PUCE), Quito 170143, Ecuador; cdsolorzano@puce.edu.ec; 2International Faculty of Innovation PUCE-Icam, Pontificia Universidad Católica del Ecuador (PUCE), Quito 170143, Ecuador

**Keywords:** soil stabilization, organic soil, nanosilica, undrained shear strength, elastic modulus

## Abstract

Organic soil is widely recognized for its low shear strength and high compressibility, which pose challenges for construction projects. One of the most commonly used methods for enhancing the mechanical properties of soil is chemical stabilization using various additives. In this study, the undrained shear strength of organic soil from Quito, Ecuador, with an average organic content of 43.84%, was reinforced using 0.5, 1, 3, and 6% nanosilica. A series of tests, including Atterberg limit, specific gravity, compaction, and unconfined compression tests, were conducted on specimens cured for 28 days. The results indicate that increasing the nanosilica content leads to higher plasticity, lower maximum dry density, and higher optimum moisture content. In addition, the modulus of elasticity and undrained shear strength improved. The optimal nanosilica content was found to be 1%, resulting in a 211.28% increase in the undrained shear strength. The mechanisms of soil improvement driven by the chemical interactions between nanosilica, mineralogical components (analyzed via XRD), and soil organic matter are discussed in detail.

## 1. Introduction

### 1.1. Background of the Improvement and Stabilization of Soils

Soil plays a crucial role in structural design, because loads are transmitted through the foundation, inducing stress and deformation in the soil [1]. In some cases, it is necessary to modify the properties of problematic soils such as soft clays, expansive clays, and organic soils [2,3]. Improving organic soils is particularly challenging because of their complex chemical interactions. However, chemical stabilization using binders such as lime, ash, or cement has been extensively studied, yielding highly satisfactory results. Despite their effectiveness, these traditional methods have been shown to contribute to an unsustainable carbon footprint over time [4,5].

Therefore, binders with nanometer-sized particles have been increasingly studied. Although they entail high initial construction costs, they significantly enhance the mechanical behavior of soils while reducing long-term maintenance expenses [6].

For soil stabilization, nanomaterials are applied at various dosages across different soil types, and are sometimes combined with other materials to enhance their mechanical properties. Rasool Haji and Ahmed Mir [7] investigated the effect of nano-gypsum at dosages ranging from 1% to 2% in combination with cement at dosages between 10% and 20%. Yao et al. [8] performed direct shear tests on silty clay treated with cement and nano-magnesia (MgO) to investigate their influence on the shear strength. At a cement content of 10%, the optimum nano-MgO content was determined to be 10%. Kannan et al. [9] employed nano-calcium carbonate in a low-plasticity soil with moderate organic content.

Nanosilica is one of the most extensively studied materials because of its colloidal particle size, which provides a high specific surface area and surface charges that promote strong chemical activity during the formation of pozzolanic compounds [10,11]. The most crucial reactions are the formation of a calcium silicate hydrate (C-S-H) gel (CaO–SiO_2_–H_2_O), which results from the reaction with Ca^2^⁺ cations [12,13].

Ghadr et al. [14] studied the behavior of nanosilica in mudstone soil with a pH range of 5 to 7, using nanosilica dosages between 0.3% and 1.2% and curing periods of 7, 14, and 28 days. The study concluded that for all dosages and curing times, the soil exhibited a decrease in the maximum dry density and an increase in the optimum moisture content after compaction. Additionally, the unconfined compressive strength and modulus of elasticity increased, with the best results observed at the 28-day curing time.

Karimiazar et al. [15] evaluated the effect of nanosilica and lime on marl soil with a pH of 8.2, using nanosilica dosages ranging from 0% to 1.2% and lime dosages from 0% to 8%, with curing periods of 7, 14, and 28 days. The study concluded that nanosilica alone tends to increase the liquid limit and plasticity index, while slightly reducing the plastic limit. Additionally, it decreased the maximum dry density, increased the optimum moisture content after compaction, enhanced the shear strength, increased the effective friction angle, and slightly improved cohesion. The results also indicate that curing time positively influences the improvement of these characteristics.

Gu et al. [16] investigated the effect of nanosilica in dosages ranging from 0% to 2% in clay soil (pH 7.92) and sandy soil (pH 9.95), both without curing and with a 28-day curing period. This study demonstrated that in both cases, although more effectively at 28 days, nanosilica reduced water evaporation over time in both soil types while increasing the unconfined compressive strength, friction angle, and cohesion. Additionally, the study evaluated pH changes after nanosilica application and found that the pH of the clay soil decreased to 7.48, whereas the pH of the sandy soil dropped to 8.74.

García et al. [17] studied the effect of nanosilica on the consistency limits of high-plasticity clayey soil from Texcoco Lake, using dosages ranging from 0% to 18% in 11 soil samples. The study concluded that the liquid limit and plasticity index generally increased as more nanosilica was added, although in some samples, these values tended to stabilize at approximately 5% nanosilica content. Conversely, the plastic limit was only slightly affected, and remained constant beyond the addition of approximately 5% nanosilica.

### 1.2. Organic Soil

Organic soils are composed of unconsolidated organic matter at various stages of decomposition and are typically formed under conditions of excessive moisture. Their physical properties, such as particle size and porosity, depend on changes in chemical composition resulting from decomposition and fossilization [17,18]. The most common precursor to the formation of organic soils is the flooding of valley plains, which deposits a layer of alluvial debris at the bottom. Combined with low temperatures and high precipitation, these conditions create an anaerobic environment that slows the decomposition of plant material, leading to peat accumulation [19,20,21].

Organic soils are characterized by their dark color, putrid odor, spongy consistency, high permeability, high water content, high void ratio, and low density, among other properties [22]. Soils with an organic content exceeding 20% are classified as organic soils, whereas those with more than 75% organic content are referred to as peat [23]. The presence of organic matter significantly influences the geotechnical properties of the soil. Huat et al. [24] conducted multiple tests on soils from Malaysia with varying organic content and concluded that as organic content increases, specific gravity and bulk density decrease, while the plasticity and in situ moisture content increase.

Organic matter in soil consists of living organisms, such as bacteria and fungi, and dead organic material at various stages of decomposition. Partially decomposed materials have a low specific surface area, contributing to a lower density of organic soils than those with minimal organic content. In contrast, fully decomposed material, known as humus, plays a crucial role in soil properties because its particles function as colloids derived from plants, animals, and microbes [24,25].

Humus formation occurs through microbial respiration, a process by which soil microorganisms break down organic matter to obtain energy from glucose. Under anaerobic conditions, respiration, also known as fermentation, leads to the decomposition of glucose into pyruvic acid, which is then further broken down into lactic acid through lactic fermentation or into ethanol through alcoholic fermentation. The glucose used in this process originates from soil organic matter [26,27]. This decomposition results in humus [20], which is primarily composed of humic acids that are soluble under weakly acidic conditions, and fulvic acids that are soluble across all pH ranges [28]. Additional components include lignin, which is resistant to microbial decomposition, and carbohydrates from undecomposed plant material. Humic compounds are considered amorphous and colloidal substances with a high cation exchange capacity. The cation exchange process is mainly driven by carbohydrates such as cellulose from organic matter, hydroxyl groups (phenolic and alcoholic), and carboxyl groups. Hydroxyl groups are more abundant in humic acids, whereas carboxyl (–COOH) groups predominate in fulvic acids. The latter significantly contributes to the total acidity of humic compounds by readily releasing protons H⁺ in acidic environments [29].

## 2. Materials and Methods

### 2.1. Sampling Site

The soil studied originates from the southern part of Quito, in the “El Garrochal” sector. The test pit is located at coordinates 0°20′23.41″ S latitude and 78°31′56.84″ W longitude, at an elevation of approximately 2990 m above sea level (m.a.s.l.) (Figure 1). The inter-Andean basin consists of tectonically controlled basins that are generally composed of volcano-sedimentary deposits of fluvial origin. The study site lies within the Quito-San Antonio-Guayllabamba Basin, near the Atacazo and Pichincha volcanoes through which the Machángara River flows [30]. More specifically, it is located in the Sanguanchi River sub-basin in the Turubamba sector [31], an area characterized by sediments such as silt, clay, medium to coarse sands, volcanic ash, and pumice [32]. According to Mayanquer et al. [33] and Calderón-Carrasco et al. [34], a prehistoric lagoon may exist in this region. Over time, this lagoon likely drained into the Machángara River, leaving behind sediments that now form the soil layers of the region.

### 2.2. Sample Extraction and Storage

The soil sample was extracted from a single test pit, starting at 0.50 m, where the soil exhibited apparent homogeneity in its composition, and extending to a depth of 1.30 m (Figure 2a). Because the tests in this study did not require undisturbed samples, soil portions were excavated down to 1.30 m (Figure 2b). The extracted soil was then manually broken up and homogenized using a shovel to minimize variations associated with depth and ensure a uniform sample (Figure 2c). Finally, the mixed sample was stored under conditions that prevented exposure to sunlight, temperature fluctuations, and humidity changes (Figure 2d).

### 2.3. Testing Process and Proposed Homogenization of Organic Content and Nanosilica

Despite the initial homogenization process, the organic content of the soil subsamples remained uneven. This inconsistency could significantly affect the mechanical properties of the soil, leading to high variability in the results. As previously mentioned, variations in the organic content directly affect the mechanical behavior of the soil. To address this issue, a second homogenization process was performed on the soil subsamples using a hand blender (Figure 3a,b).

A common issue encountered when mixing nanomaterials is agglomeration, which reduces the effectiveness of stabilization [35]. To achieve homogeneity, conventional methods involve oven-drying the soil before adding and dry-mixing the nanomaterials. Water was then introduced, followed by a second mechanical mixing [36]. However, organic soil cannot be oven-dried without altering its properties because it is prone to carbonization, oxidation, or vaporization of organic matter [37]. Another frequently employed homogenization technique involves the application of nanosilica as a liquid slurry. Although this approach preserves the soil’s physical properties, it was deemed unsuitable given a soil in situ moisture content of approximately 400%. The addition of water further prolongs the subsequent processes of room-temperature drying required for compaction testing and sample remolding for unconfined compression assessments. Both procedures require moisture contents that are markedly lower than the soil’s natural in situ level.

To prevent the alteration of soil properties, a second mechanical mixing was performed in the wet state using a hand blender for at least 15 min to ensure uniform distribution (Figure 3c,d), the dosing in this study was performed based on the dry weight of the subsamples. Given the division of the sample into multiple subsamples for various tests, triplicate controls were used to measure the moisture and organic content of each subsample. This step allowed for the following:Determination of the dry weight for accurate dosing;Evaluating the degree of dispersion of organic content across the entire sample;Assessment of the effectiveness of the mechanical homogenization process using a hand mixer.

Figure 4 summarizes the sampling, homogenization, and laboratory testing processes established for this study.

### 2.4. Nanosilica Properties

The nanosilica powder used in this study was supplied by XFNANO, a company specializing in nanomaterial manufacturing based in Nanjing, Jiangsu Province, China. Table 1 lists the key properties of the nanosilica used in this study.

Nanosilica (NS) is porous, amorphous, low-density silicon dioxide (SiO_2_) with particle sizes ranging from 1 to 100 nm [38]. When properly mixed, nanosilica disperses throughout the soil and can be considered as a colloid in a theoretically pure state. It is commonly synthesized using the sol-gel method, which allows precise control over the SiO_2_ particle size [39].

The formation of silicon dioxide (SiO_2_) particles began with the synthesis of SiO_2_ molecules. This process involves the bonding of hydroxyl OH groups from an aqueous medium to silica atoms, forming silicic acid (H_4_SiO_4_). As these molecules cluster, condensation occurs, leading to polymerization and hydrogel formation. Upon drying, the hydrogel agglomerates into spherical SiO_2_ particles, creating a surface-hydroxylated xerogel characterized by Si-OH bonds on its surface, as illustrated in Figure 5a [40].

Inside the amorphous SiO_2_ particles, three-dimensional structures with energetically unstable bonds exist, despite having complete octets. However, these octets are often incomplete on the particle surface. As a result, they tend to complete their octets by adsorbing water molecules and undergoing hydroxylation [41].

In this study, the manufacturer stated that SiO_2_ is obtained by precipitation, the sol-gel method, or the fumed-phase method. All three methodologies expose the material to high temperatures, leading to dehydroxylation of the particle surface. This dehydroxylation leaves the surface electrically negative, because the surface tetrahedra remain incomplete, resulting in a negative valence of the residual surface oxygen. If a dehydroxylated silica particle is placed in an aqueous medium, its Si-O-Si bonds undergo hydrolysis because of the dipolar nature of the water molecules, which tend to dissociate into hydroxyl OH⁻ and hydrogen H⁺ ions. These ions then bind to the Si-O-Si bonds on the dehydroxylated particle surface, as illustrated in Figure 5b. If the hydrolysis process continues, it progresses toward the core of the particle, ultimately forming H_4_SiO_4_ (silicic acid) on its surface [42].Si-OH + H⁺ ‪→ Si-OH_2_^+^(1)

In a strongly acidic medium, the solubility of SiO_2_ particles in the formation of silicic acid (H_4_SiO_4_) is slow. Some H⁺ ions in the medium interact with the exposed Si-OH bonds, forming protonated hydroxyl groups OH_2_⁺, as shown in Equation (1). This interaction positively charges the particles [43] (pp. 172–173). Although protonated hydroxyl groups OH_2_⁺ are less reactive, they can form hydrogen bonds with water molecules more easily because of the dipolar nature of water. As a result, silicic acid formed on the surface cannot condense into Si-O-Si bonds or subsequently polymerize into a rigid network [40]. Instead, silicic acid chains remain dissolved in the acidic medium, with their protonated hydroxyl groups promoting greater adsorption of water molecules through interactions between OH_2_⁺ and water [43] (pp. 172–173).

In an alkaline medium, some OH⁻ ions from the medium interact with SiO_2_ particles, which are significantly more soluble. As a result, the formation of silicic acid (H_4_SiO_4_) occurs rapidly and tends to deprotonate, forming SiO- ions that impart a negative charge to the particles, making them highly reactive, as shown in Equation (2) [43] (pp. 172–173). These silicate ions bond together to form polysilicates that can condense into capillary channel networks with Si-O-Si bonds, commonly referred to as silica gels.

The rigidity of these silica gels increases with increasing concentration, and the gelation rate is strongly influenced by both silica concentration and pH of the medium [44].Si-OH + OH^-^ ‪→ Si-O + H_2_O(2)

### 2.5. Physical and Chemical Soil Characterization

Soil characterization was conducted in triplicate, except for the compaction test, which was performed in duplicate, following the standards outlined in Table 2. All tests were carried out on wet samples to prevent alterations in soil properties due to oven-drying. However, the minimum sample quantities were controlled according to the standard requirements, with moisture content tests performed on samples dried at 60 °C, as recommended by ASTM D2216 [45] for organic soils.

Studies on the specific gravity of organic soils suggest that drying temperatures between 60 °C and 70 °C are optimal because lower temperatures (below 50 °C) can retain significant amounts of residual water in soil pores, whereas higher temperatures can degrade organic matter [46]. ASTM D854 [47] advises against the use of water for testing highly organic soils because their organic content is often less dense than the water density and may float. Instead, kerosene is typically recommended for better deaeration and prevent flotation [48]. However, in this study, wet samples were de-aerated in water and subsequently dried at 60 °C, as the soil used did not exhibit flotation despite its high organic content.

**Table 2 nanomaterials-15-00702-t002:** Summary of physical characterization laboratory tests.

Laboratory Test	Parameter	Standard	Test Realized Number
Particle-Size Distribution	Gravel, sand, lime, and clay fraction [%]	ASTM D 7928 [49] ASTM D 1140 [50]	3 for each standard
Atterberg Limits	LL, LP, IP [%]	ASTM D 4318 [51]	3 for each NS Content, for a total of 15
USCS Classification	Soil Classification	ASTM D 2487 [52]	3
Organic Content	Organic fraction [%]	ASTM D 2974 [53]	30
Fiber Content	Fiber fraction [%]	ASTM D 1997 [54]	3
Specific Gravity	Gs	ASTM D 854 [47]	3 for each NS Content
Laboratory Compaction	ɣd max., W opt.	ASTM D 1557 [55]	At least 8 points for each NS Content, for a total of 47 points
Unconfined Compressive Strength	qu, E	ASTM D 2166 [56]	3 for each NS Content, for a total of 15

Sieve and sedimentation analyses were conducted to determine the particle size distribution. The results are summarized in Table 3, and the corresponding particle size distribution curve is presented in Figure 6.

In addition to the previously mentioned tests, the pH of the soil, both in its natural state and with 6% nanosilica content after 28 days of curing, was measured using a potentiometer in an aqueous solution with a 1:2.5 ratio. The cation exchange capacity was determined using atomic absorption spectrophotometry and the electrical conductivity was measured using a conductivity meter with a saturated paste extract. Table 4 summarizes the results of these tests.

For chemical characterization, the sample was calcined in a muffle furnace for two hours at a controlled ascending temperature of 650 °C. This process allowed the determination of mineralogical compounds with defined crystallization present in the sample through X-ray diffraction (XRD). The results for the mineral components are presented in Table 5 and Figure 7 shows the XRD graph.

## 3. Results

### 3.1. Organic Content

This study does not fully represent the extensive organic soil deposits at the site, which exhibit considerable variability in their organic content. Instead, a sample representing this extent was used. The frequency histogram shown in Figure 8 is based on 30 organic content tests of the soil subsamples used in this study, as summarized in Table 6. The trend approximates a symmetrical Gaussian bell curve, and considering the sample size, it is inferred that the observed trend will not change significantly with a larger number of organic content tests but will become more refined [57].

Therefore, it can be inferred that the organic content after the proposed homogenization process follows an approximately normal distribution. The average organic content of the samples was 43.84%, the median was 43.65%, and the mode was 43%. The close proximity of these three values supports the hypothesis that the data follow a symmetric, approximately normal distribution, as in a normal distribution, these values typically coincide. The overall standard deviation was 1.99%, which indicates that there is a 95% probability that the organic content of the sample lies between 39.87% and 47.82%. Similarly, with 95% confidence, the average organic content of the sample was estimated to be between 43.10% and 44.59%. Although this study considered the average organic content to be homogeneous, it also highlighted its grade of variability, as variations in organic content significantly affect soil properties, even within the same area [24].

### 3.2. Effect of Nanosilica on Plasticity

Table 7 lists the consistency limits for each nanosilica content level. It can be observed that the liquid limit, as well as the plasticity index, show an increase of less than 2% at low dosages (ranging from 0.5% to 1%). At 3%, the increase was 10.30%, and beyond this dosage, the values stabilized. On the other hand, the plastic limit at the 6% dosage was only reduced by 6.38%. Figure 9a shows one sample tested in triplicate, while Figure 9b illustrates the graphical relationship between the consistency limits for different nanosilica contents.

### 3.3. Effect of Nanosilica on Specific Gravity

Table 8 presents the values of the specific gravity of the soil along with the standard deviation for each nanosilica content tested in triplicate, as shown in Figure 10a,b. The average specific gravity of the natural soil was 1.91. At dosages ranging from 0.5% to 3%, there was no significant variation in specific gravity. At 6%, there was a slight reduction in its value; however, this decrease was marginal, and the specific gravity remained relatively constant.

The standard deviations were generally small, indicating that the data were highly consistent across all dosages.

### 3.4. Effect of Nanosilica on Maximum Dry Density and Optimum Water Content

Based on granulometry, Method A from the ASTM standard was selected, and the sample was prepared using the wet method, meaning that the soil was allowed to dry at room temperature, away from direct sunlight, with humidity checks until the target moisture content was reached for compaction. Figure 11a,b show the equipment used and soil sample tested, respectively. Table 9 and Figure 11c present the results of the compaction test for each nanosilica content level, along with the respective coefficient of determination (R^2^), which evaluates the quality of the quadratic regression curve used to represent soil behavior. An increase in the optimal moisture content was observed; however, the maximum increase at 6% nanosilica content was 3.32%, which was very slight. On the other hand, a reduction in the maximum dry density was observed, with the maximum reduction corresponding to the 6% nanosilica content, which was 3.45%, a value that is not significant.

### 3.5. Effect of Nanosilica on Undrained Shear Strength and Elastic Modulus

The effect of nanosilica content on the undrained shear strength was evaluated using an unconfined axial compression test with strain control for each nanosilica content level. The cylindrical samples were remolded under optimum moisture and maximum dry density conditions, as shown in Table 9, with a height/diameter ratio specified in the ASTM standard referenced in Table 2. Table 10 lists the average compaction and saturation parameters of the samples. During the unconfined compression test, it was noted that the failure plane could not be distinctly identified in several specimens. Figure 12 shows a specimen prior to the execution of the test along with the irregular failure planes developed as a result of the test.

The stress–strain curves for each nanosilica content level are shown in Figure 13a–e, from which the undrained shear strength (Su) was determined as half of the maximum stress value of each plot. The secant modulus of elasticity corresponding to 50% of the maximum compressive strength (E_50_) is also derived. Table 11 and Figure 14a,b demostrate a significant increase in both the shear strength and modulus of elasticity, with the largest improvement observed at 1% nanosilica content. In addition, the standard deviations of both the parameters are presented. As these values are low, it can be inferred that the results are consistent.

## 4. Discussion

### 4.1. Role of Soil Mineralogical Components in the Improvement with Nanosilica

From granulometry, it is known that the soil consists of approximately 14.55% sand fraction and 85.45% fine fraction, which is further divided into 49% particles ranging from 0.075 to 0.002 mm (silts) and 36.45% particles smaller than 0.002 mm (clays). Of the latter, only the clay fraction that falls within the mentioned particle-size range can be classified as colloids. X-ray diffraction analysis revealed that a large portion of the crystalline components in the soil was composed of minerals from the plagioclase group. These minerals belong to the feldspar family, a subgroup of aluminosilicates primarily composed of silica, aluminum, and variable amounts of sodium and calcium. Silicon and aluminum readily combine with oxygen to form crystalline frameworks known as tectosilicates. Because of this affinity, plagioclase is the most abundant mineral in Earth’s crust [58,59]. The presence of this mineral can be attributed to the proximity of the site to the *Atacazo* and *Pichincha* volcanoes, where silica is abundant in the form of ash or lava flows that are subjected to weathering in acidic environments [43] (pp. 147–148) and [60,61].

This mineral forms stable Si-O-Si tetrahedral siloxane bonds in its structure, with silanol-Si-OH bonds present only on its surface [43] (pp. 147–148). In general, these minerals do not exhibit plasticity and are chemically inert, because tetrahedral and octahedral bonds create a very stable crystalline structure [61]. They exhibit a low capacity for volume change and, owing to the particle size mentioned earlier, have a low specific surface area. The majority of sand and silt are composed of this mineral and do not exhibit colloidal properties [43] (pp. 133–134).

Based on the methodology described, it was determined that the soil had a low pH, classifying it as an acidic medium. This condition promotes the weathering of plagioclase through the hydrolysis of its surface bonds, driven by the high concentration of H⁺ ions present in the soil. These ions break bonds on the surface of the particles. Weathering facilitates the release of cations such as Ca^2^⁺ and Mg^2^⁺, along with other metals that constitute the plagioclase group. This process is influenced by the crystalline structure of the mineral; since plagioclase has a stable structure with stronger and more uniform bonds, more energy is required to break it through hydrolysis.

From Table 3, it can be seen that the soil is predominantly composed of silt-sized particles and has a high concentration of exchangeable Ca^2^⁺ cations. This suggests that the soil underwent some degree of weathering, likely due to the acidity of the organic medium beneath the water table. However, the stable crystalline structure of minerals from the plagioclase group, along with the particle sizes present in the soil, limit the extent to which mineralogical components influence cation exchange. Therefore, the plagioclase group does not actively participate in cation exchange, but contributes to the release of Ca ^2^ ⁺ cations through the weathering process prior to the application of nanosilica.

In an alkaline environment, C-S-H gel formation is favored because the negative charge of the silica particles, provided by superficial Si-O- bonds, easily attracts Ca^2^⁺ cations to achieve electrical equilibrium. However, C-S-H gel formation in a strongly acidic environment is not favorable because of the slow hydrolysis of SiO_2_. However, SiO_2_ particles can still form H_4_SiO_4_. This allows interaction between Ca^2^⁺ ions and H_4_SiO_4_, leading to the formation of a C-S-H gel and further acidification of the medium.

Equation (3) approximates this reaction, although the exact composition of the gel may vary depending on the medium conditions.Ca^2^⁺ + H_4_SiO_4_ ‪ → C-S-H + 2H^+^(3)

Table 4 presents the pH values of the same subsample, both without nanosilica and with 6% nanosilica content, after 28 days of curing. The pH decreased from 3.79 to 3.63, which could indicate the formation of a C-S-H gel due to the release of additional H⁺ ions. This suggests that the addition of nanosilica to soil tends to reduce its pH value [16].

### 4.2. Role of Soil Organic Matter in the Improvement with Nanosilica

Table 4 indicates that the soil has a high organic content, a low pH, and contains 10.92% of its dry weight in fibers. From this, it can be inferred that humus is the predominant type of organic matter, given its low fiber content and soil acidity.

Amorphous nanosilica exhibits higher solubility and greater reactivity with organic molecules than silica-based minerals such as plagioclase found in soil [42,62,63]. Most organic colloids, including humic and fulvic acids, interact with silica through adsorption on the surface of H_4_SiO_4_, which forms via the hydrolysis of SiO_2_ particles [42,64]. This adsorption occurs through hydrogen bonding with the hydroxyl OH- and carboxyl -COOH groups [65,66].

Nanosilica has demonstrated a notably strong interaction with soil colloids, particularly organic matter colloids, exceeding that of other nanomaterials. According to Mustoe [65], silica can be deposited onto the hydroxyl functional groups of cell walls and carboxyl –COOH groups through Si-O-C and Si-C bonds [67,68,69], forming a mineral film and filling intercellular voids, thereby enhancing the structural rigidity. In addition, the time required for this process depends on the concentration of silica available for the reaction.

Figure 15 illustrates how the hydroxyl and carboxyl groups, carbohydrates, and cations interacted with the hydrolyzed nanosilica surface.

## 5. Conclusions

Nanosilica can be considered a highly effective material owing to its significant contribution to the increase in the undrained strength, as discussed in this study. Additionally, the importance of determining the optimal nanosilica content is emphasized, as excessive amounts can negatively impact the soil’s maximum strength. The key conclusions drawn from this study are as follows.

Effect on Plasticity: Nanosilica increases the plasticity of organic soil by raising its liquid limit and lowering its plastic limit. This was attributed to the high water adsorption capacity of the nanosilica, which enabled it to retain more water in its plastic state.Specific Gravity: Nanosilica had no significant effect on the specific gravity of soil. The slight variation in specific gravity values can be attributed to the variability in organic content, as mentioned previously.Maximum Dry Density: The reduction in the maximum dry density with increasing nanosilica content is due to the low density of amorphous and porous nanosilica particles [68]. However, organic soil also exhibits a low dry density owing to the presence of partially decomposed organic matter and the high water adsorption capacity of organic colloids, which results in a large percentage of their weight being water. Therefore, the slight decrease in the maximum dry density is attributed to the similar low densities of the two materials.Optimum Moisture Content: The increase in the optimum moisture content is associated with a higher specific surface area of the mixture. This implies that the nanosilica surface, through additional ionic attraction, enhances the adsorption capacity of water molecules, thereby increasing water retention [69].Undrained Shear Strength and Elastic Modulus: The undrained shear strength and modulus of elasticity of the soil increased significantly with the addition of 1% nanosilica, which was identified as the optimal content. At higher concentrations, nanosilica tends to agglomerate rather than effectively fill soil voids [70]. This increase in strength can be attributed to multiple interaction mechanisms.

C-S-H Gel Formation: Although influenced by pH, C-S-H gel can form throughout the 28-day curing process because of the availability of Ca^2^⁺ ions for cation exchange, and the high reactivity and colloidal properties of nanosilica. Additionally, evaluating the soil at the maximum dry density and optimum moisture content enables a more effective interaction between nanosilica and Ca^2^⁺ ions, preventing their dispersion into the medium owing to excess free water.Interaction with Colloids and Organic Molecules: Nanosilica interacts with colloids and organic molecules, forming multiple chemical bonds that enhance the stiffness of the soil matrix and consequently increase soil strength.Enhancement of inter-particle interactions: The compaction process densifies the soil mass by reducing the trapped air in the voids and leaving free water in these spaces. Nanosilica adsorbs free water, promoting greater contact between soil particles, which enhances internal friction and increases resistance. However, excess nanosilica can agglomerate, thereby reducing the adsorption capacity. Figure 16 illustrates this mechanism.

This study proposes an innovative approach for improving organic soils through the incorporation of nanosilica, whose application in this soil type has been previously limited because this type of soil is typically avoided in construction owing to its poor mechanical properties. The results indicated that nanosilica significantly increased the shear strength of the soil, even at low dosages. Specifically, it was found that an optimal dosage of 1% by weight relative to dry soil was sufficient to achieve significant improvements in undrained shear strength values. This finding is particularly significant because it reduces the need for large volumes of additives and enables more efficient stabilization compared to conventional methods. In conclusion, this research opens new possibilities for stabilizing organic soils through the methodology employed in this study, which integrates the behavior of both mineralogical and organic soil components in response to nanosilica through the lens of synergistic chemical mechanisms among nanosilica, organic matter, and mineral components contributing to soil improvement.

## Figures and Tables

**Figure 1 nanomaterials-15-00702-f001:**
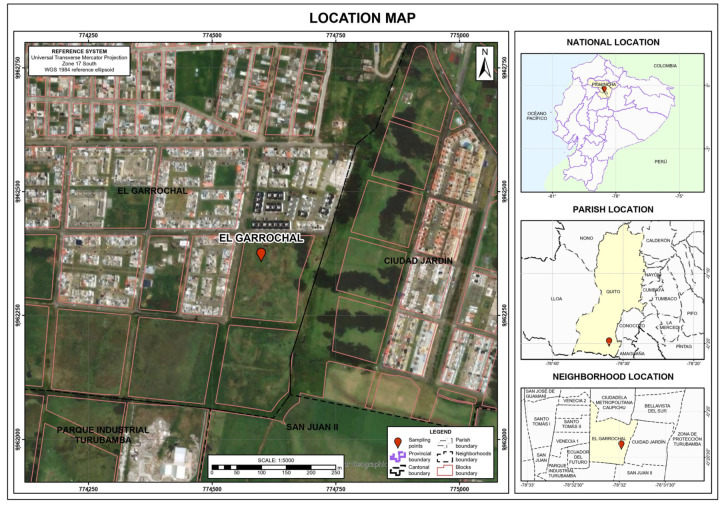
Test Pit Location.

**Figure 2 nanomaterials-15-00702-f002:**
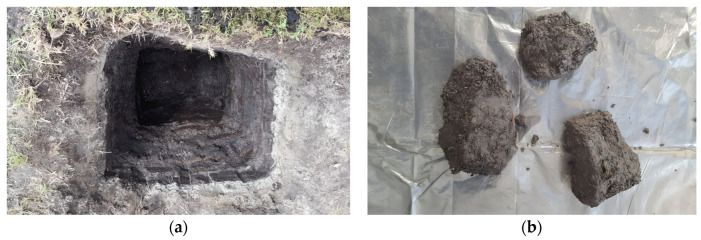
(**a**) Test pit excavation; (**b**) soil extraction from the test pit; (**c**) soil homogenization process using a shovel; (**d**) soil storage.

**Figure 3 nanomaterials-15-00702-f003:**
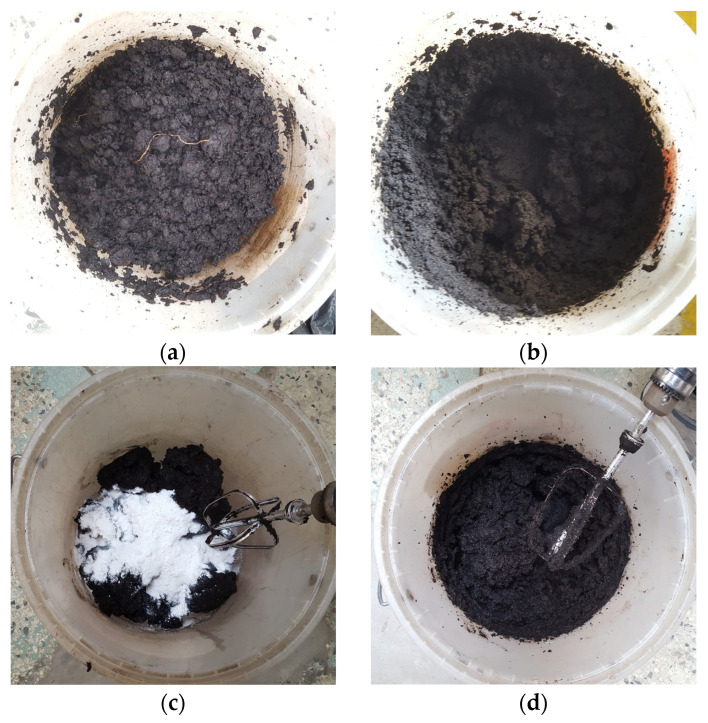
(**a**) Soil prior to homogenization of the organic content; (**b**) soil after homogenization of the organic content; (**c**) soil prior to nanosilica mixing; (**d**) soil after nanosilica mixing.

**Figure 4 nanomaterials-15-00702-f004:**
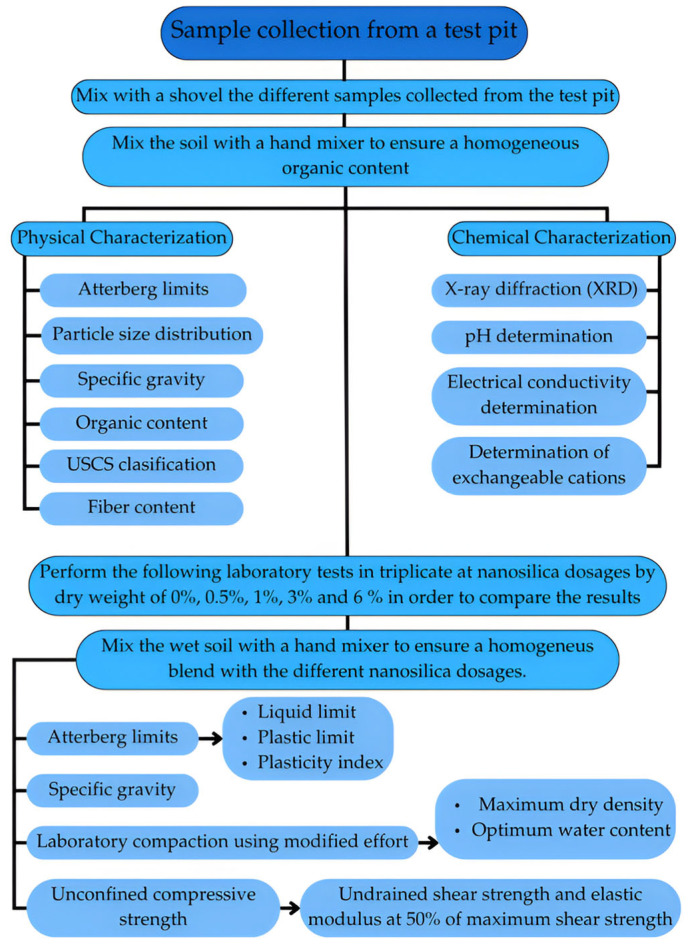
Map of the proposed methodology.

**Figure 5 nanomaterials-15-00702-f005:**
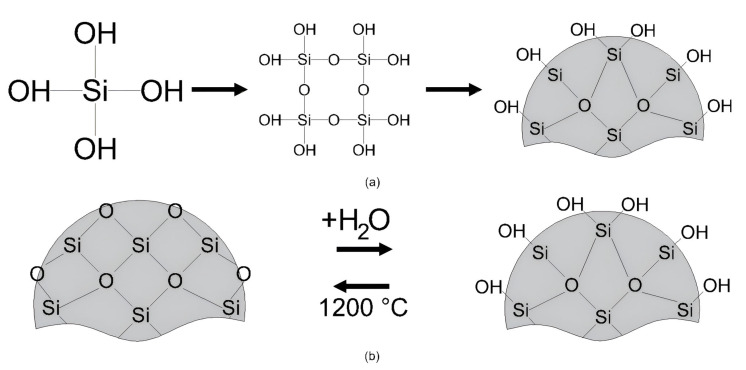
(**a**) Formation of SiO_2_ particles; (**b**) rehydroxylation process through hydration. (modified from Zhuravlev [40]).

**Figure 6 nanomaterials-15-00702-f006:**
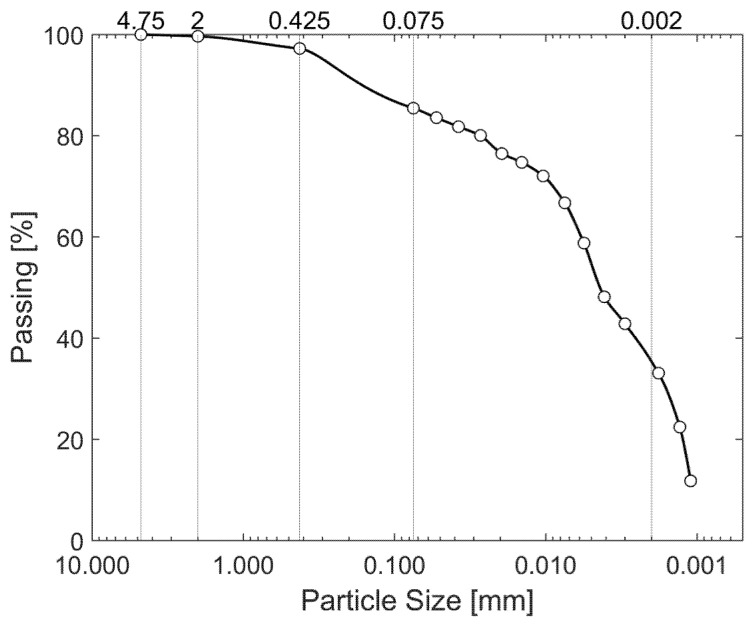
Particle size distribution of soil.

**Figure 7 nanomaterials-15-00702-f007:**
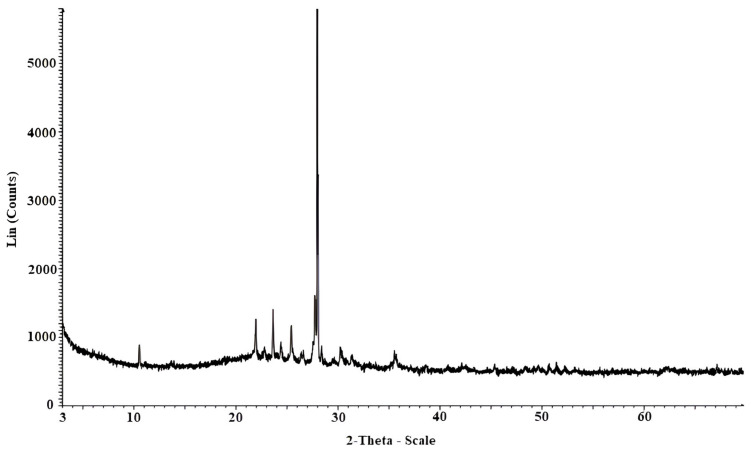
XRD analysis of organic soil.

**Figure 8 nanomaterials-15-00702-f008:**
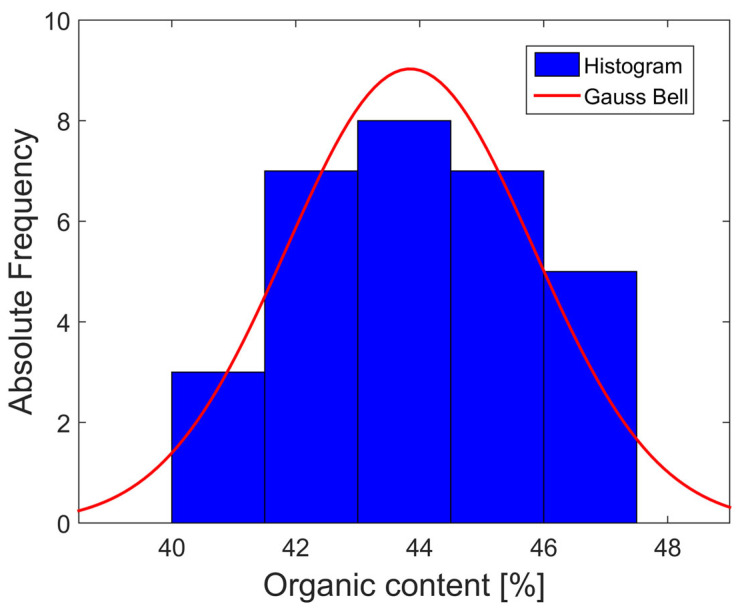
Histogram and Gauss bell of organic content.

**Figure 9 nanomaterials-15-00702-f009:**
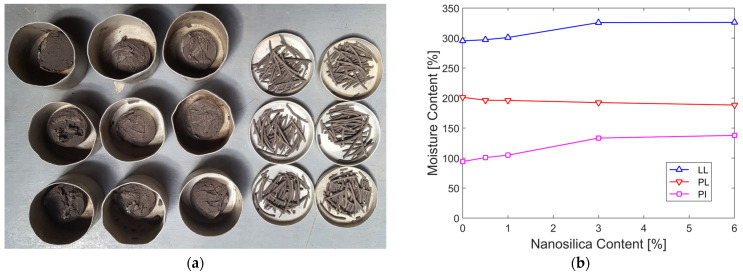
(**a**) Atterberg limits tests performed in triplicate; (**b**) effect of nanosilica on plasticity.

**Figure 10 nanomaterials-15-00702-f010:**
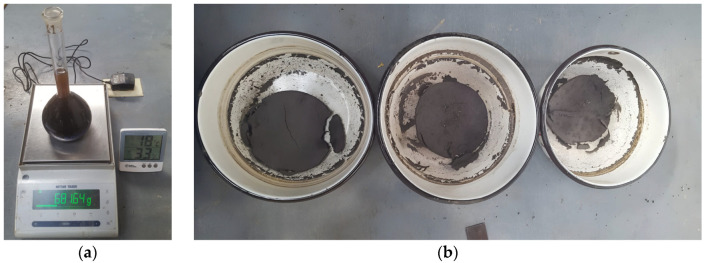
(**a**) Specific gravity test performed; (**b**) specific gravity test performed in triplicate after drying.

**Figure 11 nanomaterials-15-00702-f011:**
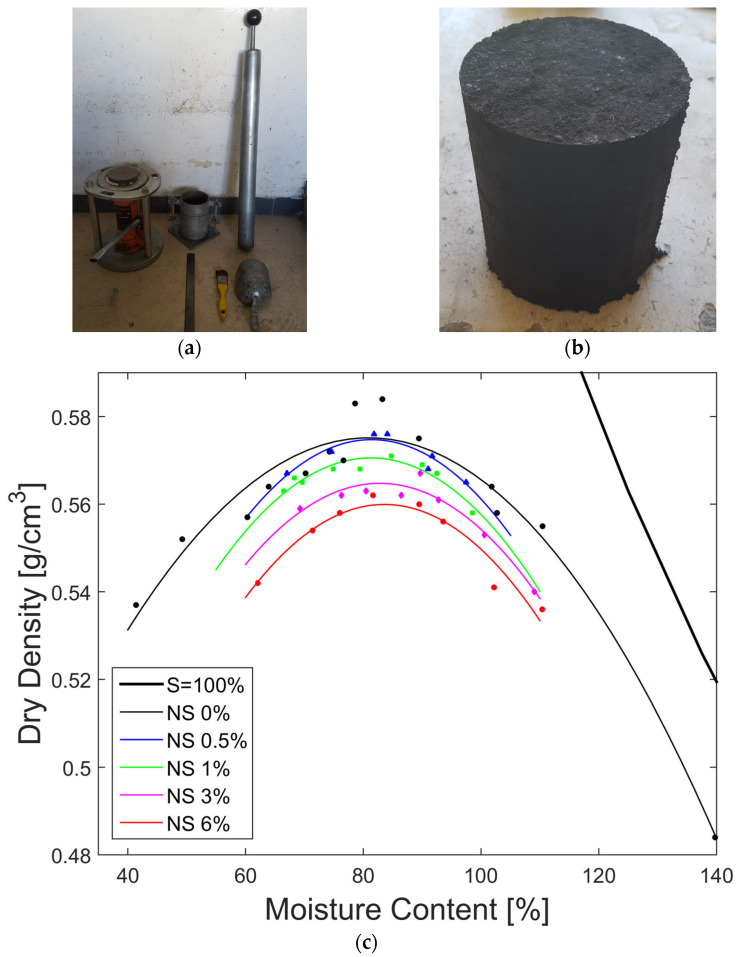
(**a**) Equipment used for the compaction; (**b**) sample removed from the mold after the compaction test; (**c**) effect of nanosilica on maximum dry density and optimum water content graph.

**Figure 12 nanomaterials-15-00702-f012:**
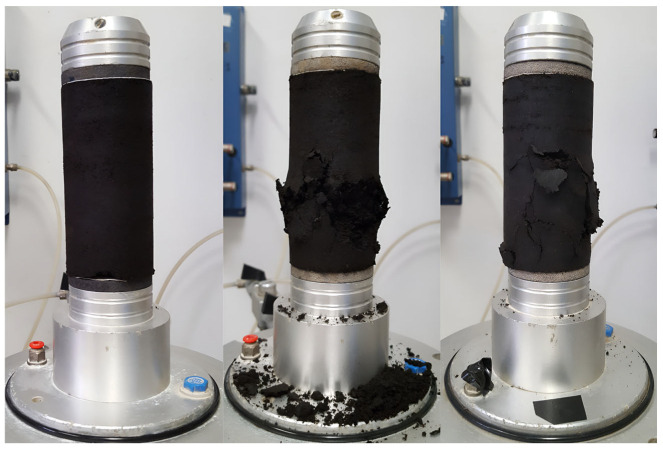
Irregular failure planes of the tested samples.

**Figure 13 nanomaterials-15-00702-f013:**
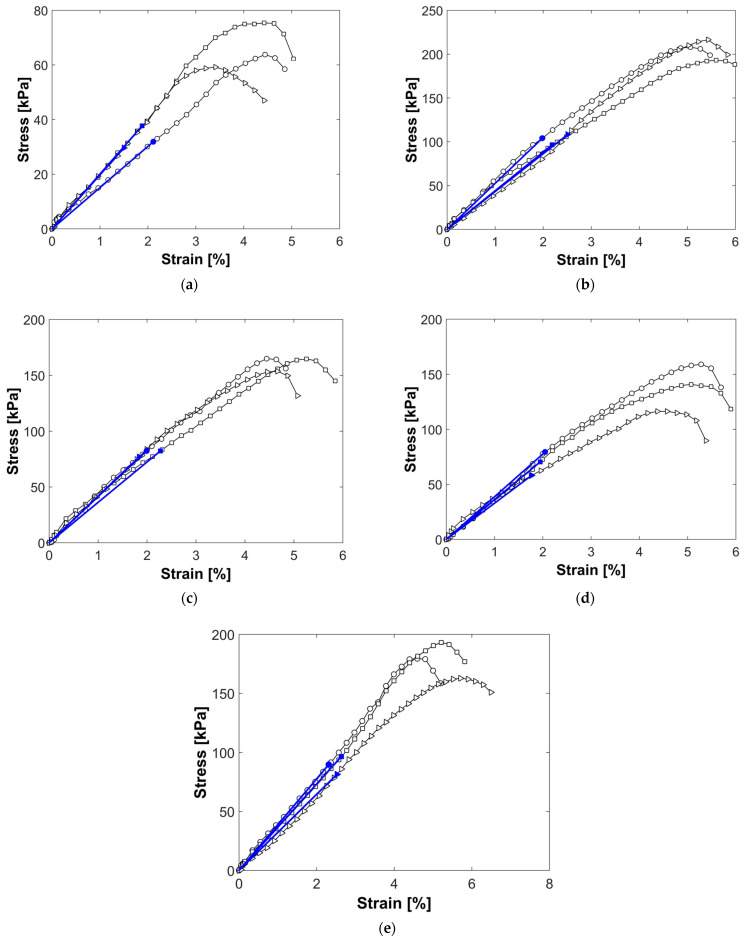
Unconfined compressive strength after 28 days of curing for nanosilica contents of (**a**) 0%; (**b**) 0.5%; (**c**) 1%; (**d**) 3%; (**e**) 6%.

**Figure 14 nanomaterials-15-00702-f014:**
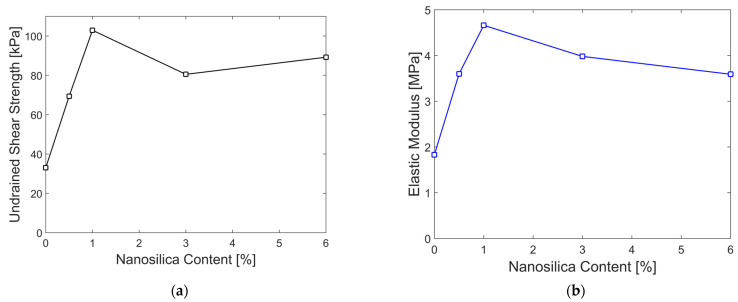
(**a**) Relationship between unconfined compressive strength and nanosilica content; (**b**) relationship between elastic modulus at 50% of the maximum compressive strength and nanosilica content.

**Figure 15 nanomaterials-15-00702-f015:**
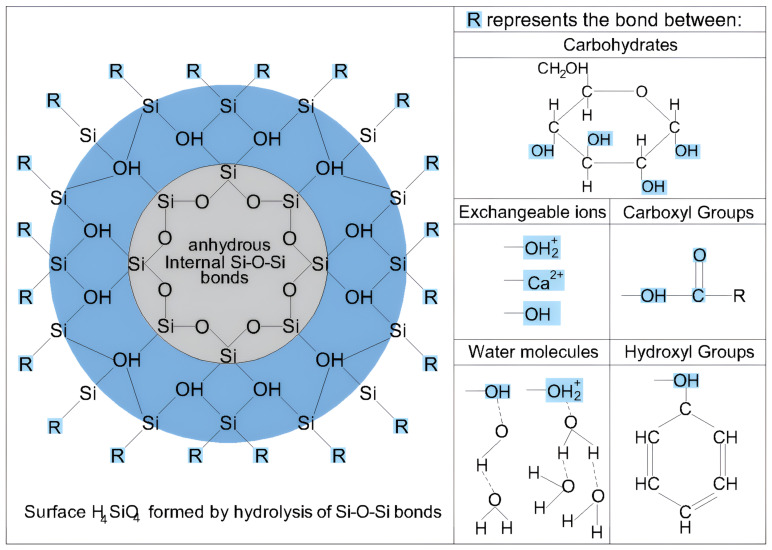
Interactions between the nanosilica surface, cations, and organic colloids.

**Figure 16 nanomaterials-15-00702-f016:**
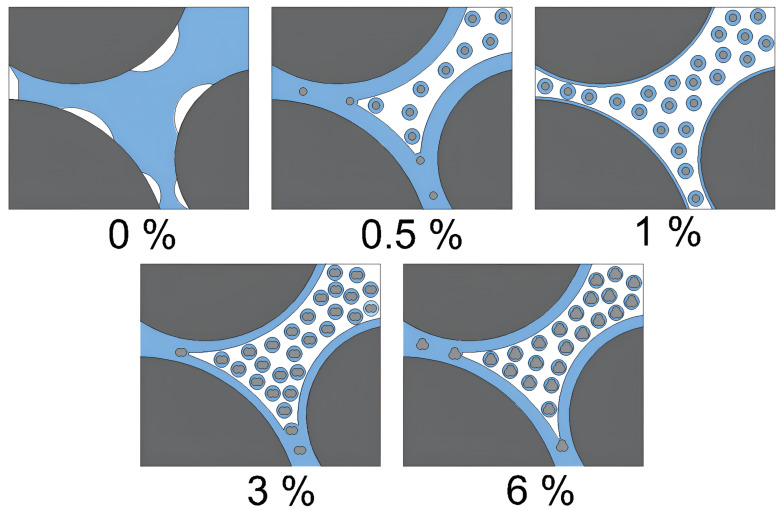
Mechanism of free water adsorption and enhancement of inter-particle interactions.

**Table 1 nanomaterials-15-00702-t001:** Properties of nanosilica.

Property	Value
Chemistry Formula	${\mathrm{SiO}}_{2}$
Morphology and Color	Amorphous white powder
Particle Size	20 nm (0.00002 mm)
Specific Surface Area	145–160 m^2^/g
Purity	99%

**Table 3 nanomaterials-15-00702-t003:** Summary of particle-size distribution tests.

Particle Size Distribution by Washing		Particle Size Distribution by Hydrometer	
Passing 4.75 mm:	100%	Particles smaller than 0.075 mm:	83.58%
Passing 2.00 mm:	99.82%	Particles smaller than 0.005 mm:	55.32%
Passing 0.475 mm:	98.84%	Particles smaller than 0.002 mm:	36.45%
Passing 0.075 mm:	85.39%		

**Table 4 nanomaterials-15-00702-t004:** Summary of the organic soil properties.

Property	Value	Property	Value
Sand fraction	14.55%	Ash content	High ash content
Silt fraction	48.99%	Fiber content	10.92%
Clay fraction	36.45%	Specific gravity	1.91
Liquid limit, LL	295.34%	pH at 0% of NS	3.79
Plastic limit, PL	201.18%	pH at 6% of NS	3.63
Plasticity index, PI	94.16%	Electric conductivity	3.71 dS/m
Activity Index	2.58	Exchangeable cation “Ca”	10.20 cmol/kg
USCS soil classification	Organic Silt (OH)	Exchangeable cation “Mg”	2.15 cmol/kg

**Table 5 nanomaterials-15-00702-t005:** Mineral composition of the soil determined by XRD.

Mineral	Chemical Formula	Content (%)
Plagioclase Group	(Na,Ca)Al(Si,Al)Si_2_O_8_	85
Anhydrite	CaSO_4_	6
Cordierite	Mg_2_Al_4_Si_5_O_18_	4
Maghemite	Fe_2_O_3_	3
Sodalite	Na_8_(AlSiO_4_)_6_(ClO_4_)_2_	2

**Table 6 nanomaterials-15-00702-t006:** Summary of organic content for each test.

Test	% Nanosilica	Organic Content	Average (Partial)	Standard Deviation (Partial)	Overall Average	Overall Standard Deviation
USCS	-------	43.66	43.63	0.60	43.84	1.99
		43.02				
		44.22				
Particle Size Distribution by Hydrometer	-------	47.01	45.35	1.65		
		45.34				
		43.70				
Specific Gravity	0	41.53	44.48	2.52		
	0.5	46.66				
	1	45.61				
	3	41.99				
	6	46.60				
Consistency Limits	0	43.34	42.71	0.84		
	0.5	43.64				
	1	41.69				
	3	42.01				
	6	42.88				
Maximum Dry Density and Optimum Water Content	0	42.41	44.54	2.01		
		40.21				
	0.5	43.13				
		46.41				
	1	44.71				
		45.54				
	3	46.41				
		45.33				
	6	45.69				
		45.55				
Unconfined Compressive Strength	0 to 6	41.03	41.76	1.45		
		42.35				
		40.19				
		43.47				

**Table 7 nanomaterials-15-00702-t007:** Summary of the effect of nanosilica on plasticity.

% Nanosilica	Average LP	Average LL	Average IP	LL Increase	LP reduction	IP Increase	LP Standard Deviation	LL Standard Deviation
0	201.18	295.34	94.16	-------	-------	-------	0.92	1.18
0.5	196.38	297.19	100.82	0.63%	2.39%	7.07%	1.41	0.51
1	196.05	300.94	104.89	1.90%	2.55%	11.40%	0.69	0.63
3	192.46	325.76	133.31	10.30%	4.33%	41.58%	1.28	1.69
6	188.35	326.12	137.77	10.42%	6.38%	46.31%	0.85	1.44

**Table 8 nanomaterials-15-00702-t008:** Summary of the effect of nanosilica on specific gravity.

% Nanosilica	GS	Average	Standard Deviation
0	1.90	1.91	0.035
	1.94		
	1.87		
0.5	1.92	1.90	0.026
	1.89		
	1.87		
1	1.91	1.92	0.015
	1.91		
	1.93		
3	1.92	1.91	0.017
	1.89		
	1.91		
6	1.87	1.88	0.020
	1.86		
	1.90		

**Table 9 nanomaterials-15-00702-t009:** Summary of the impact of nanosilica on ɣd max, W opt.

% Nanosilica	Maximum Dry Density (g/cm^3^)	Optimum Water Content (%)	R^2^	Max Dry Density Reduction	Optimum Water Content Increase
0	0.58	80.92	0.96	-------	-------
0.5	0.57	81.40	0.86	1.72%	0.59%
1	0.57	81.84	0.86	1.72%	1.14%
3	0.56	82.81	0.95	3.45%	2.34%
6	0.56	83.61	0.93	3.45%	3.32%

**Table 10 nanomaterials-15-00702-t010:** Summary of average compaction and saturation parameters of the samples.

% Nanosilica	Average Water Content [%]	Average Dry Density [g/cm^3^]	Average Void Ratio	Average Degree of Saturation [%]
0	80.41	0.58	2.30	66.86
0.5	80.71	0.57	2.32	66.08
1	82.38	0.57	2.36	67.03
3	83.22	0.56	2.38	66.74
6	85.33	0.56	2.38	67.50

**Table 11 nanomaterials-15-00702-t011:** Summary of the impact of nanosilica on shear strength and elastic modulus.

% Nanosilica	Undrained Shear Strength (Su) [kPa]	Elastic Modulus (E_50_) [MPa]	(Su) Standard Deviation	(E_50_) Standard Deviation	Su Increase [%]	E_50_ Increase [%]
0	33.06	1.83	4.17	0.28	-------	-------
0.5	69.36	3.60	10.68	0.30	109.80	96.72
1	102.91	4.66	5.91	0.51	211.28	154.64
3	80.52	3.98	3.20	0.32	143.56	117.49
6	89.20	3.59	7.61	0.33	169.81	96.17

## Data Availability

The analyzed data can be provided upon request.

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
