# Peer review of "Effect of Nanosilica on the Undrained Shear Strength of Organic Soil"

_nanomaterials, 2025, doi:10.3390/nano15090702_

Round 1

Reviewer 1 Report

Comments and Suggestions for Authors

The manuscript (entitled: Effect of Nanosilica on the Undrained Shear Strength of Organic Soil) pays main attention on the undrained shear strength of organic soil from Quito by adding Nanosilica. Their work indicate that increasing the nanosilica content leads to higher plasticity, lower maximum dry density, and higher optimum moisture content. This work is interesting and meaningful. Yet, I still think the authors should improve the manuscript in content. My detail comments are listed as following points.

1) The content of the “Introduction” can be simplified partly.

2) The figure 5 should be revised.

3) In the section 3, the authors carried out detail tests. Yet, can the shear face of failure face be observed?

4) I think the CT test can be used to find the micro- inner failure of Organic Soil and Nanosilica- Organic Soil, as similar case in reference “Application of computerized tomographic scanning to the study of water-induced weakening of mudstone”.

5) The conclusion section should be revised to highlight the innovative work and findings.

Reviewer 2 Report

Comments and Suggestions for Authors

No, I consider this MS is fine to be accepted.

The location map could be showing clearly and the experiment methods could be described more detailedly. How many samples have been carried out that should be listed in a table.  

Reviewer 3 Report

Comments and Suggestions for Authors

This is a very interesting paper on the use of nanosilica for improving the physical properties of soils for construction bases. I do have some comments
1) Please include some information on the diversity of nanospecies that can be includes as soil improvement amendments. What other nanomaterials can function in this role? Why has nano silica been favored in this study.
2) You use mechanical mixing to add nanosilica to homogenized soil samples. Is this the expected method for the application of this material? How would other application methods such as the application of nano silica in a liquid slurry affect the physical properties of treated soil? Can you talk about this in the discussion?

Round 2

Reviewer 1 Report

Comments and Suggestions for Authors

This revision is acceptable.

Comments on the Quality of English Language

This revision is acceptable.